



**A multivariate statistical method for susceptibility analysis of the debris flow**
**in Southwest China**
Feng Ji[1], Zili Dai[2,*]
[1]State Key Laboratory of Geohazard Prevention and Geoenvironment Protection, Chengdu University of
Technology, Chengdu 610059, China.
[2]Center for Natural Disaster Reduction Research and Education, Shimane University, 1060 Nishikawatsu-cho,
Matsue, Shimane 690-8504, Japan.
*Corresponding Author: 87zili.dai@gmail.com.
**Abstract**: Southwest China is characterized by many steep mountains and deep valleys due to the uplift activity
of the Tibetan Plateau. The 2008 Wenchuan Earthquake left large amounts of loose materials in this area, making
it a severe disaster zone in terms of debris flow. Susceptibility is a significant factor of debris flow for evaluating
its formation and impact. Therefore, it is in urgent need to analyze the susceptibility of debris flows in this area.
At present, the susceptibility analysis models of the debris flow in Southwest China is mainly based on
qualitative methods. Little quantitative prediction model is found in the literatures. This study evaluates 70
typical debris flow gullies as statistical samples, which are distributed along the Brahmaputra River, Nujiang
River, Yalong River, Dadu River, and Ming River respectively. Nine indexes are chosen to construct a factor
index system and then to evaluate the susceptibility of debris flow. They are the catchment area, longitudinal
grade, average gradient of the slope on both sides of the gully, catchment morphology, valley slope orientation,
loose material reserves, location of the main loose material, antecedent precipitation, and rainfall intensity. Then,
an empirical model based on the quantification theory type I is established for the susceptibility prediction of
debris flows in Southwest China. Finally, 10 debris flow gullies on the upstream of the Dadu River are analyzed
to verify the reliability of the proposed model. The results show that the accuracy of the statistical model is 90%.
**Keywords**: Debris flow, susceptibility, prediction model, factor index system, multivariate statistical method.

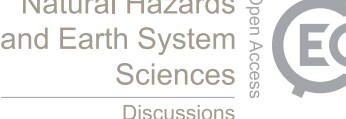
## 1 Introduction

Debris flows are a common geological hazard in mountainous areas, which transport large amounts of sediment down-slope and cause serious damage to dwellings, roads, and other lifelines. China has mainly mountainous topography and is one of the most debris-flow prone countries in the world. Until March 2019, there are approximately 50,000 debris flows have occurred in China (Di et al. 2019). A significant percentage of these debris flows are distributed in Southwest China, particularly in the Wenchuan earthquake area, where large amounts of loose material were produced by the earthquake-induced landslides (Huang et al. 2015; Dai et al. 2017).

Due to the complex nature of debris flows, it is quite difficult to fully understand their initiation mechanism and precisely forecast their occurrence (Takahashi and Das, 2014). The uncertainty of debris flows poses significant threats to human lives in downstream areas (Schürch et al. 2011). Debris flow susceptibility expresses the occurrence possibility of debris flow in an area with respect to its geomorphologic characteristics (Kappes et al. 2011; Bertrand et al. 2013). Therefore, susceptibility analysis is an essential step to conduct the risk assessment of debris flow hazards. (Di et al. 2019; Zou et al. 2019).

Debris flow susceptibility analyses include two steps: 1) identification of the potential source areas and 2) prediction of the possible deposition areas (Kang and Lee, 2018). In the literature, a large number of prediction models have been proposed for the susceptibility analyses of debris flows. For the first step, statistical models that use various environmental factors contributing to possible instabilities are well-established. For example, Guinau et al. (2007) used a bivariate statistical procedure to carry out a terrain failure susceptibility analysis on debris flows that occurred in Nicaragua. Blahut et al. (2010) performed susceptibility assessment for the source areas of landslide induced debris flows in the Valtellina Valley based on bivariate statistics. Bertrand et al. (2013) performed two multivariate statistical models, a linear discriminant analysis (LDA) and a logistic regression (LR), to analyze the debris flow susceptibility of upland catchments. Jomelli et al. (2015) proposed a Bayesian hierarchical probabilistic model to investigate how debris flows respond to environmental and climatic variables in the French Alps. Carrara et al. (2008) discussed the application of different statistical models to debris flows in Val di Fassa, Trento Province. Lucà et al. (2011) compare bivariate and multivariate statistical models for the evaluation of gullying susceptibility in Northern Calabria, South Italy, and concluded that multivariate statistical



models were found to be the best model in predicting debris flow susceptibility of the study area. For the second
step, the concept "angle of reach" was widely used in the empirical models to predict the runout distance of the
debris flows (Scheidl and Rickenmann, 2010; Hürlimann et al. 2012; Horton et al. 2013). Recently, many
numerical models were proposed to simulate the propagation of the debris flows and predict the deposition area.
For example, Pirulli and Sorbino (2008) analyzed the propagation of potential debris flows in Southern Italy
using two numerical codes RASH3D and FLO2D. Beguería et al. (2009) proposed a two-dimensional model
based on numerical integration of the depth-averaged motion equations to predict the debris flow propagation
over complex terrain near Lienz, Eastern Tyrol, Austria. Huang et al. (2015) presented a numerical model based
on the smoothed particle hydrodynamic (SPH) method to calculate the runout distance of catastrophic debris
flows that occurred in the Wenchuan Earthquake area. Gregoretti et al. (2016) used a cell model to simulate a
debris flow that occurred on the Rio Lazer. Moradi et al. (2017) performed debris flow susceptibility zoning of
debris flows in the Province of Reggio Calabria based on the SPH method. Some recent analysis methods of
debris-flow susceptibility could be found in Cama et al. (2017), Prieto et al. (2018), and Rosatti et al. (2018).
The previous studies mentioned above have attempted to conduct debris flow susceptibility analysis in specified
regions. Southwest China is characterized by steep mountains and deep valleys, and is strongly affected by the
uplift activity of the Tibetan Plateau. Moreover, Southwest China has abundant loose material after the 2008
Wenchuan Earthquake. Therefore, a series of large-scale debris flows have been occurred during the rainy
seasons in Southwestern China (Wu et al. 2019). At present, the susceptibility analysis of the debris flow in this
area is mainly investigated based on qualitative methods with relevant specifications (Xu et al. 2013; Di et al.
2019). This work aims at providing a multivariate statistical method for susceptibility analysis of the debris flow
in Southwest China. 70 debris flow gullies in Southwest China were analyzed, and nine key indicators were
extracted through the initial analysis of the debris flows. Through multivariate statistics, an empirical formula of
susceptibility was established, which was then validated with 10 debris flow gullies on the upstream of the Dadu
River. It is worth noting that this work confines to identify the potential debris-flow source areas in Southwestern
China, neglecting the runout of the phenomena.



## 2 Study area characteristics of debris flow

Southwest China is charactered by steep mountains and deep valleys and is strongly affected by the uplift activity of the Qinghai–Tibet Plateau. Furthermore, there is abundant loose material and rainfall in this area. Therefore, it is a severe disaster zone in terms of debris flow. In the past three years, 70 typical debris flows distributed along the Brahmaputra River, Nujiang River, Yalong River, Dadu River, and Ming River are investigated. The location of the debris flows is shown in Figure 1. The formation condition of these debris flows in deep valley zones are analyzed, and a prediction assessment model for debris flow susceptibility is established based on a multivariate statistical method. The characteristics of the research area are summarized as follows.

In the upstream of the Brahmaputra River, 18 debris flows along the Dagu River and Jiexu River reaches are investigated. The lithology in this area is the irruptive rock of the late Yanshanian–Himalayan epoch, with a wide distribution of granodiorite. The average annual rainfall in this area is about 540 mm and concentrates mostly in summer. Large-scale ice-melting-type debris flow once occurred in this region. However, in recent years, the debris flows in this area are mainly caused by precipitation. Material reserves are abundant in the valleys, whereas unstable materials are found less frequently and the deposit zone is small. It is found that most of the debris flows in this area are in the decline phase, and most debris flow gullies are in the low-frequency category.

In the midstream of the Nujiang River catchment, 11 debris flow gullies located in the Zuogong River section are investigated. The stratum mainly includes the Permian Nacuo group slate and Triassic Wapu group marble. As this region is located in the subtropical zone south of the Himalayas, it is characterized by a humid climate and plentiful precipitation. This leads to an extensive distribution of debris flow gullies. In the midstream of the Yalong River catchment, 27 debris flow valleys are investigated, which belong to a plateau climate zone with complex meteorological and hydrological conditions. The concentricity and suddenness of rainfall provide hydraulic conditions for the debris flow breakouts. Collapses and landslides in the valley occur frequently. Moreover, the debris flow activity is intensified by unreasonable human engineering activities such as deforestation and accumulation of highway waste residues.

In the Dadu River catchment, 42 gullies in the midstream and the upstream are surveyed. This area is characterized by intense new tectonic movement, high earthquake intensity, and rock fragmentation on the mountain surface. Debris flow, collapse, and other geological disasters are widely distributed, and the deposit



zone of the debris flow is large. The maturity of the valley is high.
In the Minjiang River catchment, the Wenchuan River section are surveyed, and 32 debris flows are investigated.
This region is characterized by abundant loose materials, frequent debris flows, and high possibility of the
breakout of large-scale debris flows. Most of these debris flows are intensive in activity, occur very frequently,
and have not declined in recent times.
**3 Methodology**
*3.1 Investigation and statistical data*
In total 70 debris flow gullies distributed in five water catchments in Southwestern China are investigated from
the gully outlet to the watershed over the past three years. This work includes the investigation of the watershed
terrain, geological structure, outbreak scale, loose material distribution, processes of occurrence and movement,
frequency of debris flows, and so on. The role of each factor causing instability of the source materials are
investigated. In addition, the precipitation data before the outbreak of debris flows are collected from local
meteorological bureaus. The impulse force, sediment discharge, and other dynamic parameters are calculated.
*3.2 Field test*
All of the 70 debris flow gullies are traced, and bulk density tests (size 50 cm × 50 cm × 50 cm) and
screening tests of the loose material are conducted on the deposit zone to determine the composition of
debris flow sources. Besides, according to the superelevation and flow depth of the curved gully zone, the
speed of the debris flow is estimated to provide the basic data for the dynamic parameter calculation.
*3.3 Drilling and geophysical prospecting*
For the active debris flow gullies, the geologic condition is complex. Considerable resources are invested
in drilling and geophysical prospecting to obtain the volume, material composition, structure, and content
of the fine-grained soil precisely.
*3.4 Statistical technique*
The statistical techniques can be grouped into bivariate and multivariate methods. A bivariate statistical





method analyses each parameter individually, therefore the calculation and application in bivariate
statistical models are straightforward and efficient (Suzen and Doyuran, 2004). On the other hand, a
multivariate statistical method considers the interaction of all parameters in controlling the occurrence of a
phenomenon, and is considered as one of the best methods in predicting debris flow susceptibility (Lucà et al.
2011). Hayashi's quantification theory is a well-known multivariate statistical method developed by Hayashi
(1961). It is widely used in various fields, such as risk assessment (Zhang et al. 2003; Jiang et al. 2010),
psychological analysis (Sato et al. 1994), sociological surveys (Li et al. 2011; Han, 2014), and financial
statistics (Choi et al. 2009). In this method, the qualitative and quantitative variables could be mutually
transformed based on a reasonable principle. Therefore, this method has very good applicability to process
the quantitative and qualitative influencing factors of debris flow risk.
Qualitative variables are termed items in quantification theory. All possibilities for each item are termed
categories. A dummy variable $\delta_i(j, k)$ is introduced in the model to express the response of an item and the
category for each sample:
$$\delta_1(j,k) = \begin{cases} 1, & \text{if response of } i\text{th sample in the category } k \text{ of} \\ & \text{item } j \text{ to the corresponding external criterion;} \\ 0, & \text{otherwise.} \end{cases} \quad \begin{cases} \text{i} = 1, 2, \ldots, \text{ n;} \\ \text{j} = 1, 2, \ldots, \text{ m.} \end{cases} \quad (1)$$

where $n$ is the number of samples and $m$ denotes the number of items.
The response matrix $X$ can be expressed as a $n \times p$-order matrix composed of all categories $\delta_i(j, k)$:
$$X = \begin{pmatrix} \delta_1(1,1)\cdots\delta_1(1,r_1) & \delta_1(2,1)\cdots\delta_1(2,r_2)\cdots\delta_1(m,1)\cdots\delta_1(m,r_m) \\ \delta_2(1,1)\cdots\delta_2(1,r_1) & \delta_2(2,1)\cdots\delta_2(2,r_2)\cdots\delta_2(m,1)\cdots\delta_2(m,r_m) \\ \vdots \qquad \vdots \qquad \vdots \qquad \vdots \qquad \vdots \qquad \vdots \\ \delta_n(1,1)\cdots\delta_n(1,r_1) & \delta_n(2,1)\cdots\delta_n(2,r_2)\cdots\delta_n(m,1)\cdots\delta_n(m,r_m) \end{pmatrix} \quad (2)$$

To establish a quantitative analysis model, the qualitative and quantitative in-situ observations are used to
fit the linear relationship between the concerned independent variable and the dependent variable. In the
Hayashi's quantification theory, the random variable changes with the $m$ variables:
$$y_i = \sum_{j=1}^{m} \sum_{k=1}^{r_j} \delta_i(j,k) b_{jk} + \varepsilon_i, \quad i = 1, 2, ..., n \quad (3)$$





where $y_i$ represents the susceptibility of the $i$th debris flow gully. $r_j$ is the number of categories of the item
$j$. $b_{jk}$ is a constant coefficient depending on category $k$ in item $j$. $\varepsilon_i$ is a random error.
To establish an analysis model of debris flow susceptibility, some necessary steps should be followed based
on Hayashi's quantification theory: 1) building an index system; 2) selecting samples and assigning values;
3) establishing the analysis model using single slopes; 4) conducting a significance test of the regression
equation and each variable, 5) applying this analysis model to regional debris flow hazards evaluation.
**4 Model generation and results**
*4.1 Indexes and categories in the statistical model*
Considering the debris flow features and index-acquisition conditions, nine indexes are selected in this work
to evaluate the susceptibility of debris flow gullies in Southwestern China, as listed in Table 1. They are
the catchment area, longitudinal grade, average gradient of slope on both sides of the gully, catchment
morphology, valley slope orientation, loose material reserves, loose material position, antecedent
precipitation, and $H_{1p}$ rainfall intensity. Each factor is classified into certain categories according to the
values shown in Table 2.
*4.2 Sample quantification*
debris flow gullies in Southwest China are selected as the sample to evaluate the performance of the
statistical model. The detail information of these debris flow gullies is listed in Table 3. The values of the
samples are assigned according to Eq. 1, and the response from each category is obtained. The sample data
then can be transformed into a "0-1" reflection matrix.
*4.3 Statistical model based on Hayashi's quantification theory*
When the quantitative theory and regression analysis take the binary-state variables 0 and 1, the equation
can be revised as the following linear regression expression:
$$y_i = a_0 + \sum_{j=1}^{f} a_j x_{ij} + \varepsilon_i \quad (i = 1, ..., n) \tag{4}$$

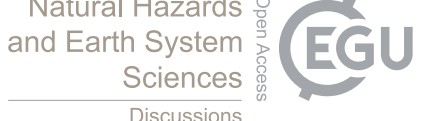



Based on Eq. 4 and matrix derivation regression calculation, the contribution values of each item are
obtained, as shown in Table 4.
Substituting the numerical values in Eq. 4, the susceptibility prediction model of debris flow is established,
which can be represented as follows:
$$
\begin{aligned}
Y = {} & 0.573x_{11} + 0.821x_{12} + 0.910x_{13} + 0.875x_{21} + 0.955x_{22} + 0.320x_{23} \\
& - 0.107x_{32} - 0.163x_{41} + 0.135x_{42} + 0.213x_{43} - 0.136x_{51} - 0.174x_{52} \\
& + 0.246x_{62} + 0.454x_{63} - 0.220x_{71} - 0.161x_{72} + 0.034x_{82} + 0.071x_{83} \\
& - 0.038x_{91} + 0.043x_{92}
\end{aligned}
\tag{5}
$$

In Eq. 5, $Y$ is the susceptibility for the debris flow. In the proposed model, the susceptibility values are
classified into three categories. When the predicted value ($Y$) is less than 1.5, the susceptibility of the debris
flow is considered as low. When $Y$ is greater than or equal to 1.5 but less than 2.5, the susceptibility is
medium. When $Y$ is greater than or equal to 2.5, the susceptibility is high. The meanings of $x_{11}$, $x_{12}$, $x_{13}$ and
other indexes are detailed in Table 2 and 3.

## 182   5 Validation and discussion

### 183   5.1 Fitting degree analysis

Table 5 shows the regression coefficient of determination and the standard deviation. As the susceptibility
of the debris flow is controlled by many factors, the coefficient of determination reaches 74.9%, reflecting
a favorable level of fit.

### 187   5.2 Self-test coincidence rate

The values of each index are used in the established model to calculate the predicted values of the
susceptibility, and then the predicted values are compared with the actual values (Fig. 2).
As shown in Fig. 2, the predicted values of debris-flow susceptibility are graded. When the predicted value
($Y$) is less than 1.5, the susceptibility to debris flow is low. When the predicted value ($Y$) is greater than or
equal to 1.5 but less than 2.5, the susceptibility is medium. When the predicted value ($Y$) is greater than or
equal to 2.5, the susceptibility is high.
From the prediction results (Table 6), the coincidence rate is 78.53% for low-susceptibility debris flow





valleys, 92.38% for medium-susceptibility debris flow valleys, 82.01% for high-susceptibility debris flow
valleys, and 86.38% for all the samples, which indicates that the regression model can predict the debris-
flow susceptibility well.
*5.3 Residual error analysis*
Figure 3 is a residual error distribution chart. It shows that the residual error fluctuates between ±0.45,
which indicates that the regression line fits the observed values well. The residual error frequency
approximates a normal distribution.
*5.4 Verification of proposed model*
The Kaka basin is located on the upper part of the Dadu River, southeast of the Qinghai–Tibet Plateau. The
valley is deep and the river runs from north to south. The regional topography is characterized by high
altitudes in the east and low altitudes in the west. The terrain is composed of high mountains with elevations
of 2000 m. There are three layers of wide valley mesas, and the uplift of mountains and river erosion is
significant. The river elevation in the Kaka basin is approximately 1800 m, the river width is 140–185 m,
and the slope angle is approximately 45–60°. The main fractures are denoted as $F_1$, $F_5$, $F_{5-1}$, $F_6$, and $F_7$ in
Fig. 4. The trend is NW, and they have a 40–60° angle with the river. A series of debris flow gullies have
occurred in the basin.
10 typical debris flow gullies on the upstream of the Dadu River are selected as samples for the model validation
(as shown in Fig. 5, and listed in Tab. 7). The accuracy of the established model is verified through the
comparison with field investigation results. Table 8 provides the relevant basic data for the samples. Each
secondary index is transformed into a 0-1 mode, and all the samples are adopted to construct a $9 \times 26$ matrix.
For the 10 verification samples, according to calculation results, the accuracy rate of the model is 90% (Tab. 8),
indicating that the prediction model is applicable to the data.
**6 Conclusions**
Debris flows frequently occurred in Southwest China and resulted in severe damage to dwellings and lifelines.
Based on the Hayashi's quantification theory, an initiation susceptibility model of debris flows in Southwest



China was proposed in this work. The following conclusions can be drawn:

1)  According to the topography and geomorphology characteristics in Southwest China, the following nine indexes were used as evaluation factors of debris flow initiation susceptibility: the catchment area, longitudinal grade, average gradient of the slope on both sides of the gully, catchment morphology, valley slope orientation, loose material reserves, location of the main loose material, antecedent precipitation, and rainfall intensity.

2)  70 typical debris flow gullies distributed along the Brahmaputra River, Nujiang River, Yalong River, Dadu River, and Ming River were investigated as statistical samples. The parameters of the prediction model were obtained based on the Hayashi's quantitative theory and regression analysis.

3)  The proposed model was applied to analyze the initiation susceptibility of 10 debris flow gullies located on the upstream of the Dadu River, and the result showed that the judgment coincidence rate is 90%, indicating that the proposed model can accurately predict the initiation susceptibility of debris flow gullies in Southwest China.

## Acknowledgments:

The presented work was supported by the Sichuan Science and Technology Program (2018JY0471), and Sichuan Provincial Youth Science and Technology Innovation Team Special Projects of China (No. 2017TD0018), the Open Fund of Key Laboratory of Geological Hazards on Three Gorges Reservoir Area (China Three Gorges University) (2018KDZ01), Ministry of Education, and the JSPS Grant-in-Aid for Early Career Scientists (19K14804).

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



**List of Table Captions**

Table 1 Nine indexes used in the prediction model of debris flow susceptibility

Table 2 Assessment index system of the model and relative categories

Table 3 Sample data for debris flow examples from Southwest China

Table 4 Score values of each index after normalization

Table 5 Quantitative model eigenvalue

Table 6 Prediction model accuracy

Table 7 Sample data from Kaka area in the upstream of Dadu River

Table 8 Comparison of predicted values and actual measured values



Table 1 Nine indexes used in the prediction model of debris flow susceptibility

| Symbol | Physical significance |
|--------|------------------------|
| $x_1$ | Catchment area ($km^2$) |
| $x_2$ | Longitudinal grade(‰) |
| $x_3$ | Average gradient of slope on both sides of gully (°) |
| $x_4$ | Catchment morphology |
| $x_5$ | Valley slope orientation |
| $x_6$ | Loose Material reserves ($10^4\,m^3/km^2$) |
| $x_7$ | Main loose material position |
| $x_8$ | Antecedent precipitation |
| $x_9$ | $H_{1p}$ rainfall intensity (mm) |




Table 2 Assessment index system of the model and relative categories

| Item | Category | value | Item | Category | Value |
|---|---|---|---|---|---|
| Catchment area $x_1$ (km$^2$) | $x_{11}$ | <1 km$^2$ | Valley slope orientation $x_5$ | $x_{51}$ | Sunny slope |
| | $x_{12}$ | 1–10 km$^2$ | | $x_{52}$ | Shady slope |
| | $x_{13}$ | 10–100 km$^2$ | Loose material reserves $x_6$ (10$^4$ m$^3$/km$^2$) | $x_{61}$ | <1 × 10$^4$ m$^3$/km$^2$ |
| | $x_{14}$ | ≥100 km$^2$ | | $x_{62}$ | 1–5 × 10$^4$ m$^3$/km$^2$ |
| Longitudinal grade $x_2$ (‰) | $x_{21}$ | <100‰ | | $x_{63}$ | ≥ 5×10$^4$ m$^3$/km$^2$ |
| | $x_{22}$ | 100‰–300‰ | Main loose material position $x_7$ | $x_{71}$ | Upstream or tributary |
| | $x_{23}$ | ≥300‰ | | $x_{72}$ | Middle and lower reaches |
| Average gradient of slope on both sides of gully $x_3$ (°) | $x_{31}$ | <30 | | $x_{73}$ | Toe of gully |
| | $x_{32}$ | 30–40° | Antecedent precipitation $x_8$ | $x_{81}$ | Inadequacy |
| | $x_{33}$ | ≥40° | | $x_{82}$ | Middle |
| Catchment morphology $x_4$ | $x_{41}$ | $Z < 0.3$ | | $x_{83}$ | Middle |
| | $x_{42}$ | $Z = 0.3$–0.7 | $H_{1p}$ rainfall intensity $x_9$ (mm) | $x_{91}$ | < 30 mm |
| | $x_{43}$ | $Z ≥ 0.7$ | | $x_{92}$ | ≥ 30 mm |

Note: $Z$ is the length to width ratio of the di basin




Table 3 Sample data for debris flow examples from Southwest China

| No. | $x_1$ | $x_2$ | $x_3$ | $x_4$ | $x_5$ | $x_6$ | $x_7$ | $x_8$ | $x_9$ | Susceptibility |
|---|---|---|---|---|---|---|---|---|---|---|
| 1 | 0.77 | 567 | 35 | Long strip | SE | 8.05 | Upstream | Inadequacy | 26.38 | Low |
| 2 | 13.3 | 366 | 28 | Ellipse | SE | 10.04 | Upstream | Inadequacy | 26.38 | Medium |
| 3 | 2.62 | 624 | 37 | Long strip | SE | 4.39 | Upstream | Inadequacy | 26.38 | Low |
| 4 | 2.47 | 624 | 36 | Long strip | SE | 26.06 | Middle and lower reaches | Inadequacy | 26.38 | Low |
| 5 | 71.64 | 194 | 22 | Ellipse | S | 8.06 | Upstream | Inadequacy | 26.38 | Medium |
| 6 | 18.89 | 344 | 35 | Suborbicular | NE | 3.08 | Upstream | Inadequacy | 26.38 | Low |
| 7 | 13.01 | 404 | 36 | Ellipse | NW | 3.43 | Upstream | Inadequacy | 26.38 | Low |
| 8 | 43.51 | 199 | 28 | Suborbicular | NE | 4.01 | Upstream | Inadequacy | 26.38 | Medium |
| 9 | 38.4 | 251 | 37 | Long strip | SE | 5.38 | Upstream | Inadequacy | 26.38 | Medium |
| 10 | 4.04 | 412.53 | 37 | Long strip | NE | 6.15 | Upstream | Inadequacy | 26.38 | Low |
| 11 | 1.39 | 480 | 35 | Long strip | N | 7.85 | Upstream | Inadequacy | 26.38 | Low |
| 12 | 1.62 | 569.4 | 36 | Long strip | S | 19.11 | Middle and lower reaches | Inadequacy | 26.38 | Low |
| 13 | 13.23 | 280.61 | 31 | Ellipse | N | 3.07 | Middle and lower reaches | Inadequacy | 26.38 | Medium |
| 14 | 2.48 | 536.68 | 41 | Long strip | S | 22.63 | Upstream | Inadequacy | 26.38 | Low |
| 15 | 5.15 | 507.69 | 39 | Ellipse | S | 10.74 | Upstream | Inadequacy | 26.38 | Low |
| 16 | 1.25 | 630.34 | 43 | Suborbicular | NE | 6.44 | Middle and lower reaches | Inadequacy | 26.38 | Low |
| 17 | 135.6 | 139.46 | 30 | Suborbicular | NE | 3.91 | Upstream | Inadequacy | 26.38 | Low |
| 18 | 53.42 | 169.87 | 30 | Ellipse | SW | 1.89 | Middle and lower reaches | Fully | 32.85 | Medium |
| 19 | 169.72 | 121.62 | 25 | Ellipse | S | 0.98 | Branch trench、Upstream | Fully | 32.85 | Medium |
| 20 | 15.53 | 171.2 | 36 | Long strip | N | 3.24 | Upstream | Fully | 32.85 | Low |
| 21 | 31.35 | 171 | 33 | Ellipse | NE | 2.74 | Middle and lower reaches | Fully | 32.85 | High |
| 22 | 7.37 | 462.11 | 35 | Suborbicular | NE | 7.06 | Middle and lower reaches | Fully | 32.85 | High |
| 23 | 20.99 | 235.79 | 25 | Ellipse | SW | 1.47 | Upstream | Fully | 32.85 | Low |



| No. | $x_1$ | $x_2$ | $x_3$ | $x_4$ | $x_5$ | $x_6$ | $x_7$ | $x_8$ | $x_9$ | Susceptibility |
|---|---|---|---|---|---|---|---|---|---|---|
| 24 | 275.41 | 60 | 23 | Ellipse | SE | 0.89 | Upstream | Fully | 32.85 | Low |
| 25 | 211.4 | 94 | 34 | Ellipse | NW | 1.04 | Tributary | Middle | 32.85 | Low |
| 26 | 8.89 | 256 | 36 | Long strip | SW | 3.79 | Upstream | Fully | 32.85 | Low |
| 27 | 28.91 | 190 | 31 | Ellipse | SE | 2.20 | Middle and lower reaches | Fully | 32.85 | Medium |
| 28 | 34.84 | 158 | 43 | Long strip | SW | 0.90 | Middle and lower reaches | Fully | 42.2 | Medium |
| 29 | 102.7 | 110 | 29 | Long strip | NE | 0.75 | Middle and lower reaches | Fully | 42.2 | Low |
| 30 | 84.81 | 146.2 | 32 | Ellipse | NE | 0.78 | Branch trench | Fully | 42.2 | Low |
| 31 | 132.02 | 129.5 | 35 | Ellipse | SW | 0.42 | Upstream | Fully | 42.2 | Medium |
| 32 | 5.5 | 318.01 | 33 | Ellipse | NE | 6.37 | Middle and lower reaches | Fully | 42.2 | Medium |
| 33 | 124.3 | 117.9 | 26 | Ellipse | SW | 1.37 | Branch trench, Upstream | Fully | 42.2 | Low |
| 34 | 26.2 | 203.9 | 36 | Ellipse | SE | 3.85 | Upstream | Fully | 42.2 | Medium |
| 35 | 29.56 | 205.1 | 32 | Long strip | SW | 1.84 | Upstream | Fully | 42.2 | Low |
| 36 | 80.34 | 119.1 | 38 | Long strip | NE | 1.51 | Branch trench, Upstream | Fully | 42.2 | Low |
| 37 | 8.45 | 301.5 | 37 | Ellipse | NE | 2.06 | Upstream | Fully | 42.2 | Medium |
| 38 | 16.26 | 217.1 | 36 | Long strip | SE | 1.15 | Branch trench, Upstream | Fully | 42.2 | Low |
| 39 | 77.5 | 138.5 | 41 | Long strip | NE | 1.22 | Upstream | Fully | 42.2 | Low |
| 40 | 23.1 | 235.52 | 24 | Long strip | SW | 1.68 | Upstream | Fully | 42.2 | Low |
| 41 | 47.01 | 166 | 30 | Ellipse | NE | 1.69 | Toe of gully | Fully | 42.2 | Medium |
| 42 | 83.11 | 125 | 31 | Ellipse | NE | 0.40 | Upstream | Fully | 42.2 | Low |
| 43 | 21.11 | 238 | 32 | Ellipse | SW | 0.87 | Upstream | Fully | 42.2 | Medium |
| 44 | 73.11 | 156 | 32 | Ellipse | SE | 1.10 | Middle and lower reaches | Middle | 43.12 | Medium |
| 45 | 64.7 | 144 | 33 | Ellipse | NW | 0.78 | Toe of gully | Middle | 43.12 | High |
| 46 | 21.87 | 242.95 | 36 | Ellipse | NW | 1.55 | Branch trench, Upstream | Middle | 43.12 | Low |
| 47 | 3.5 | 530.4 | 42 | Ellipse | NW | 8.34 | Middle and lower reaches | Middle | 43.12 | Medium |



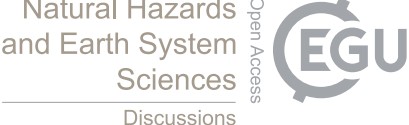
| No. | $x_1$ | $x_2$ | $x_3$ | $x_4$ | $x_5$ | $x_6$ | $x_7$ | $x_8$ | $x_9$ | Susceptibility |
|-----|-------|-------|-------|-------|-------|-------|-------|-------|-------|----------------|
| 48 | 26.66 | 296.6 | 33 | Ellipse | SE | 4.70 | Middle and lower reaches | Middle | 43.12 | Medium |
| 49 | 32.23 | 178.35 | 30 | Suborbicular | S | 5.91 | Middle and lower reaches | Middle | 28.47 | Medium |
| 50 | 40.03 | 164.6 | 31 | Ellipse | SE | 5.59 | Middle and lower reaches | Middle | 28.47 | Medium |
| 51 | 3.25 | 235.43 | 35 | Ellipse | NW | 2.18 | Upstream | Middle | 28.47 | Low |
| 52 | 351.2 | 92.4 | 24 | Ellipse | S | 10.37 | Branch trench | Middle | 28.47 | Medium |
| 53 | 8.85 | 220.35 | 36 | Suborbicular | NW | 4.01 | Branch trench, Upstream | Middle | 28.47 | Low |
| 54 | 25.31 | 203.62 | 30 | Long strip | S | 4.75 | Middle and lower reaches | Middle | 28.47 | High |
| 55 | 1.78 | 214.58 | 28 | Suborbicular | NE | 0.73 | Upstream | Middle | 28.47 | Low |
| 56 | 5.8 | 246.48 | 34 | Ellipse | SW | 15.79 | Middle and lower reaches | Middle | 28.47 | High |
| 57 | 7.6 | 230.09 | 42 | Ellipse | S | 17.34 | Middle and lower reaches | Middle | 28.47 | Medium |
| 58 | 1.7 | 140.37 | 36 | Long strip | SE | 136.82 | Middle and lower reaches | Middle | 28.47 | Medium |
| 59 | 53.27 | 132.43 | 32 | Ellipse | SE | 10.33 | Upstream | Middle | 28.47 | Medium |
| 60 | 14.15 | 178.6 | 28 | Suborbicular | SW | 55.50 | Middle and lower reaches | Fully | 41.1 | High |
| 61 | 1.48 | 244.2 | 33 | Suborbicular | SW | 32.81 | Middle and lower reaches | Fully | 41.1 | High |
| 62 | 0.89 | 256.8 | 38 | Suborbicular | SW | 18.81 | Middle and lower reaches | Fully | 41.1 | Medium |
| 63 | 0.98 | 243.2 | 35 | Suborbicular | SW | 12.70 | Middle and lower reaches | Fully | 41.1 | Medium |
| 64 | 3.73 | 120 | 24 | Long strip | SW | 9.51 | Upstream | Fully | 41.1 | Medium |
| 65 | 3.37 | 450.9 | 40 | Ellipse | SE | 8.80 | Upstream | Fully | 41.1 | Medium |
| 66 | 0.57 | 207.7 | 31 | Suborbicular | SW | 36.89 | Middle and lower reaches | Fully | 41.1 | High |
| 67 | 3.02 | 488.8 | 42 | Ellipse | SE | 20.99 | Middle and lower reaches | Fully | 41.1 | High |
| 68 | 7.59 | 352 | 28 | Ellipse | NE | 19.26 | Middle and lower reaches | Fully | 41.1 | High |
| 69 | 32.04 | 223 | 23 | Ellipse | NW | 13.67 | Middle and lower reaches | Fully | 41.1 | High |
| 70 | 3.27 | 235 | 35 | Ellipse | NE | 9.29 | Upstream | Fully | 41.1 | Low |




Table 4 Score values of each index after normalization

| Item | Category | Value | Item | Category | Value |
|---|---|---|---|---|---|
| Catchment area $x_1$ (km$^2$) | $x_{11}$ | 0.573 | Valley slope orientation $x_5$ | $x_{51}$ | -0.136 |
| | $x_{12}$ | 0.821 | | $x_{52}$ | -0.174 |
| | $x_{13}$ | 0.910 | Loose material reserves $x_6$ (10$^4$ m$^3$/km$^2$) | $x_{61}$ | 0 |
| | $x_{14}$ | 0 | | $x_{62}$ | 0.246 |
| Longitudinal grade $x_2$ (‰) | $x_{21}$ | 0.875 | | $x_{63}$ | 0.454 |
| | $x_{22}$ | 0.955 | Main loose material position $x_7$ | $x_{71}$ | -0.220 |
| | $x_{23}$ | 0.320 | | $x_{72}$ | -0.161 |
| Average gradient of slope on both sides of gully $x_3$ (°) | $x_{31}$ | 0 | | $x_{73}$ | 0 |
| | $x_{32}$ | -0.107 | Antecedent precipitation $x_8$ | $x_{81}$ | 0 |
| | $x_{33}$ | 0 | | $x_{82}$ | 0.034 |
| Catchment morphology $x_4$ | $x_{41}$ | -0.163 | | $x_{83}$ | 0.071 |
| | $x_{42}$ | 0.135 | $H_{1p}$ rainfall intensity $x_9$ (mm) | $x_{91}$ | -0.038 |
| | $x_{43}$ | 0.213 | | $x_{92}$ | 0.043 |



Table 5 Quantitative model eigenvalue

| Model | $R$ | $R^2$ | Standard deviation |
|---|---|---|---|
| 1 | 0.865 | 0.749 | 0.289 |





Table 6 Prediction model accuracy

| Category | Low | Medium | High |
|---|---|---|---|
| **Accuracy** (%) | 78.53 | 92.38 | 82.01 |



Table 7 Sample data from Kaka area in the upstream of Dadu River

| No. | Ditch name | $x_1$ | $x_2$ | $x_3$ | $x_4$ | $x_5$ | $x_6$ | $x_7$ | $x_8$ | $x_9$ |
|---|---|---|---|---|---|---|---|---|---|---|
| 1 | Luotuo | 227.1 | 102 | 25 | 0.745 | SE | 0.87 | Middle and lower | Fully | 43.8 |
| 2 | Qiongshan | 84.90 | 200 | 28 | 0.907 | SE | 10.67 | Middle and lower | Fully | 43.8 |
| 3 | Shuikazi | 49.78 | 209 | 31 | 0.534 | SE | 4.82 | Middle and lower | Fully | 43.8 |
| 4 | Bawang | 11.84 | 310 | 32 | 0.219 | SW | 2.36 | Upstream | Middle | 43.8 |
| 5 | Shenluo | 4.54 | 455 | 33 | 0.580 | NW | 42.46 | Toe of gully | Middle | 43.8 |
| 6 | Mueryue | 35.81 | 206 | 36 | 0.376 | NW | 10.08 | Upstream | Fully | 43.8 |
| 7 | Sezu | 4.23 | 613 | 42 | 0.812 | NW | 26.24 | Middle and lower | Fully | 43.8 |
| 8 | Muerluo | 11.93 | 358 | 34 | 0.546 | NW | 9.98 | Upstream | Middle | 43.8 |
| 9 | Yaneryan | 30.01 | 242 | 34 | 0.382 | SW | 5.64 | Middle and lower | Middle | 43.8 |
| 10 | Linong | 10.09 | 332 | 35 | 0.448 | NW | 24.30 | Middle and lower | Middle | 43.8 |



Table 8 Comparison of predicted values and actual measured values

| Number | 1 | 2 | 3 | 4 | 5 | 6 | 7 | 8 | 9 | 10 |
|---|---|---|---|---|---|---|---|---|---|---|
| Calculated Y value | 2.562 | 1.805 | 1.764 | 2.540 | 2.748 | 2.167 | 1.705 | 1.843 | 1.348 | 2.421 |
| Predicted susceptibility | High | Medium | Medium | High | High | Medium | Medium | Medium | Low | Medium |
| Geological judgment of actual susceptibility | High | Medium | Medium | High | High | Medium | Medium | Medium | Low | High |
| Accuracy | Right | Right | Right | Right | Right | Right | Right | Right | Right | **Wrong** |




**List of Figure Captions**

Fig.1 Distribution of the investigated debris flow gullies

Fig.2 Comparison of measured and predicted values

Fig.3 Residual distribution model of self-test standard value of susceptibility degree

Fig.4 Kaka basin geomorphology of Dadu River

Fig.5 Debris flow gullies on both sides of Dadu River




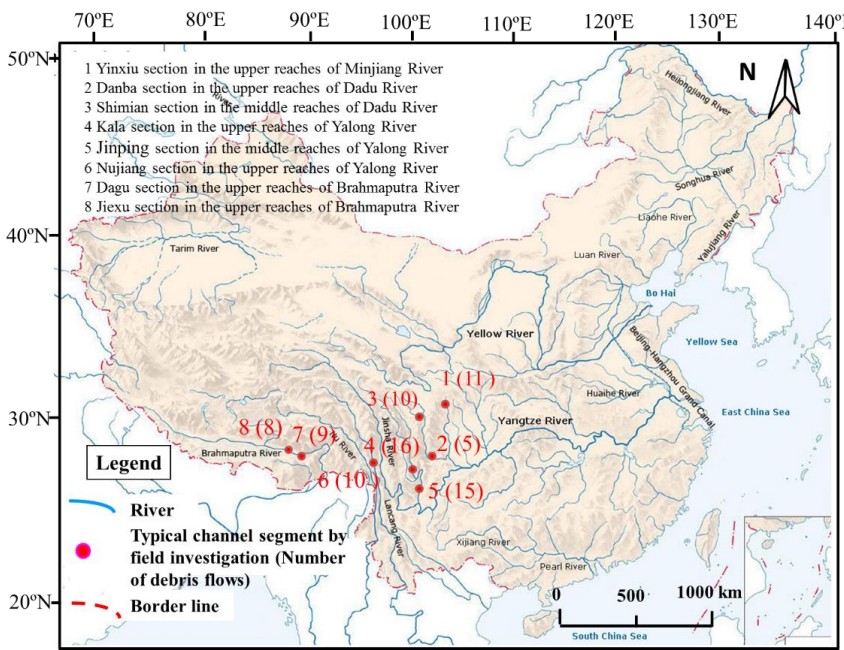

Fig.1 Distribution of the investigated debris flow gullies (the base map is from Zhao 2014)




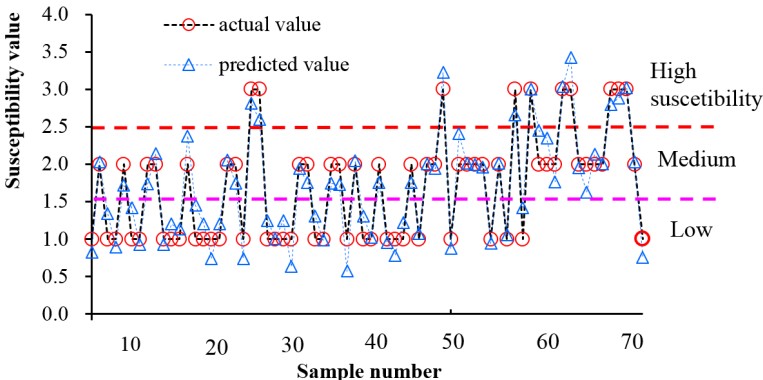

Fig.2 Comparison of measured and predicted values


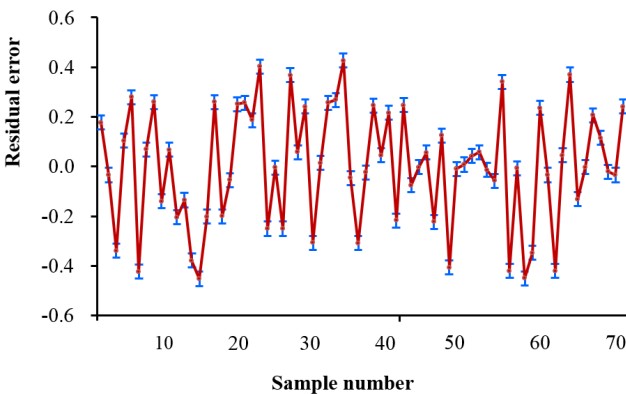

Fig.3 Residual distribution model of self-test standard value of susceptibility degree



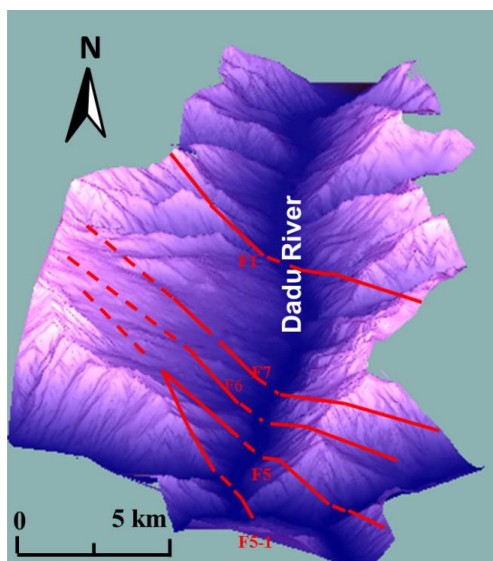

Fig.4 Kaka basin geomorphology of Dadu River


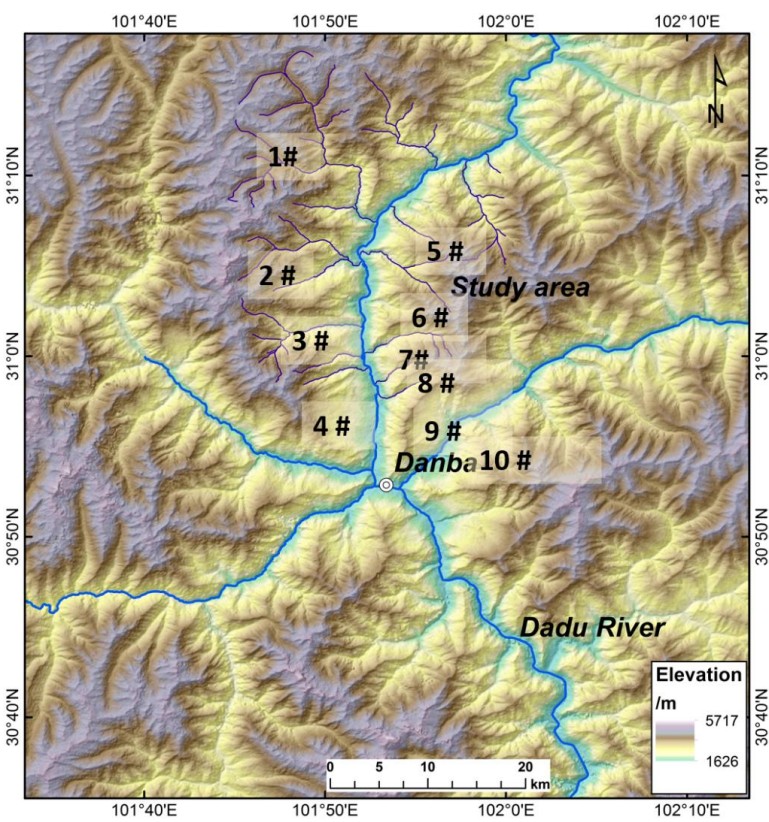

Fig.5 Debris flow gullies on both sides of Dadu River