# Peer review of "A multivariate statistical method for susceptibility analysis of the debris flow in Southwest China"

_Natural Hazards and Earth System Sciences, 2019_

## Referee Comment (RC1) · Anonymous Referee #1 · 23 Dec 2019

This manuscript presents a statistical model to analyze the susceptibility of the debris flows based on investigation data on 70 typical debris flow gullies in Southwest China. Hayashi's quantification theory was used to establish the multivariate statistical model, and nine indexes were chosen to construct the factor index system to evaluate the susceptibility of debris flows. The reliability of the proposed model was verified by the susceptibility analysis of the 10 debris flow gullies located on the upstream of the Dadu River.

Some comments and suggestions are listed as follows:

1. Geological drillings are conducted in the active debris flow gullies. The detail information about the drillings conducted in this work should be provided, such as the drilling location, the drilling results. 2. Table 2 lists nine assessment indexes used in

[Figure]

the proposed statistical model. The reason why to select these indexes to evaluate the susceptibility of debris flow gullies should be clarified. 3. In Table 2, the value of antecedent precipitation x83 should be "Fully" rather than "Middle". How to define the antecedent precipitation is "Inadequacy, Middle, or Fully"? 4. The results of the field tests mentioned in Section 3.2 should be provided and discussed. 5. In the Section 5.4, 10 debris flow gullies in the Kaka basin were analyzed to verify the accuracy of the prediction model. Please analyze the reasons why the prediction result of the Linong Gully does not match the actual susceptibility. 6. The quality of Figure 2 and 3 should be improved. For example, the horizontal axis is not correct.

---

## Referee Comment (RC2) · Anonymous Referee #2 · 10 Jan 2020

General comments: This manuscript presents a multivariate statistical model to predict the susceptibility of the debris flows in Southwest China. According to the topography and geomorphology characteristics in the study area, nine indexes were used to construct a factor index system of the statistical model. Then, 70 typical debris flow gullies in the study area were investigated as statistical samples to generate the model. 10 debris flow gullies on the upstream of the Dadu River were analyzed to verify the reliability of the statistical model. The results showed that the model has a satisfied prediction accuracy. In general, the topic of the manuscript is interesting, the methodology, results and conclusions are presented in a clear way. I recommend publication of the paper after addressing the following comments.

Specific comments:

[Figure]

1. Section 2 describes the study areas of this work. Some pictures of the typical debris flow in this area are suggested to be provided. 2. As stated in the Methodology section, the authors carried out a series of bulk density tests, screening tests, drilling and geophysical prospecting. This part should be described in more detail, and the results of these tests should be provided. 3. Line 178-180: how was the criterion determined? why 1.5? 4. Line 188-189: How to obtain the "actual values"? Please give more detail. 5. Section 5 shows the validation of the model. This part should be presented in detail. For example, what do the R and R2 mean in Table 5? How to calculate the self-test coincidence rate in Table 6? How to define the residual error in Figure 3? 6. The susceptibilities of 10 typical debris flow gullies on the upstream of the Dadu River are calculated to verify the proposed statistical model. It is suggested to show some pictures about these debris flow sites. 7. The Fig.4 should be more informative, or it is suggested to be combined with Fig.5. Besides, the quality of the figures should be improved, such as Figure 2, Figure 3, and Figure 4.

---

## Referee Comment (RC3) · Anonymous Referee #3 · 15 Jan 2020

This paper introduces an empirical model for the susceptibility prediction of debris flows in Southwest China. Nine indexes are chosen to construct a factor index system and to evaluate the susceptibility of debris flow. With the modelïijŇ70 typical debris flow gullies distributed along the Brahmaputra River, Nujiang River, Yalong River, Dadu River, and Ming River as statistical samples are assessed respectively. 10 debris flow gullies on the upstream of the Dadu River are applied to verify the reliability of the proposed modelïijŇwhich suggest a high accuracy of 90% for the statistical model. The paper is general well organized and based on plenty of investigated information. However, there are still many unclear and inexact expressions. The detail comments are as below: 1. In abstractïijŇline 14-15ïijŇ The statement"At present, the susceptibility analysis models of the debris flow in Southwest China is mainly based on qualitative

methods. Little quantitative prediction model is found in the literatures. " is not true. There have been many research work in the area after "the shock". The author should refer and comment the previous study objectively. 2. In abstractïijŇline 21ïijŇwhat is "the quantification theory type I", it never explained in the content. 3. Line 18-19 and those then after, "longitudinal grade" and "valley slope orientation" are not exact the meaning, maybe "longitudinal gradient" and "valley orientation" 4. Line 77, "Study area characteristics of debris flow", not clearly expressed, maybe "Characteristics of the debris flow in the study area" 5. Line 122, in 3.3, it just mentioned "Considerable resources are invested in drilling and geophysical prospecting", but there is no any more information and results provided. 6. Line 178-180 and 190-193, Same meaning reappears in very close distance, the sentence is also tedious. 7. Line 209, "trend" is not a professional expression, should be "strike".

---

## Author Comment (AC1) · 11 Mar 2020

**Response to the comments of Anonymous Referee #1 for nhess-2019-349**

**Answers to Technical items for which revision is required** ⸱⸱⸱ '*A multivariate statistical method for susceptibility analysis of the debris flow in Southwest China*'

The authors are grateful for the reviewer's comments and suggestions. The manuscript has been revised and each point of the reviewer's comments has been incorporated and addressed. Your comments have greatly improved the quality of this paper and we hope the revised manuscript will be of suitable standard to be accepted for publication in your journal. The main corrections in the manuscript and the responses to the reviewer's comments are as follows:

**1. Geological drilling was conducted in the active debris flow gullies. The detail information about the drillings conducted in this work should be provided, such as the drilling location, the drilling results.**

Answer: Thanks for your suggestion. To find out the material composition and the thickness of the deposit area, the geological drilling was conducted in the active debris flow gullies along the Dadu River, Yalong River, Yaluzangbo River, and Minjiang River. The drilling information and the corresponding soil characteristics are provided in the manuscript:

[revised manuscript text omitted]

**2. Table 2 lists nine assessment indexes used in the proposed statistical model. The reason why to select these indexes to evaluate the susceptibility of debris flow gullies should be clarified.**

Answer: We agree with this comment. The reason to select these nine indexes has been explained in the manuscript as:

"*There are many factors that affect the debris flow formation and development. From the perspective of source material of the debris flows, the main influence factors are catchment area, loose material position and loose material reserves. The antecedent precipitation and H1p rainfall intensity are the main generate conditions of debris flows. Besides, the catchment morphology, longitudinal gradient, average gradient of slope on both sides of the gully, and valley orientation are the main*

*factors to affect the development of debris flows. Therefore, the above nine indexes (listed in Table 3) are selected in this study to assess the susceptibility of debris flows.*" (Page 8, Lines 179-185)

**3. In Table 2, the value of antecedent precipitation x₈₃ should be "Fully" rather than "Middle". How to define the antecedent precipitation is "Inadequacy, Middle, or Fully"?**

Answer: Thank you for this comment. The classification standard of the antecedent precipitation is explained in the manuscript as:

"*The antecedent precipitation can reduce the soil shear strength, and has an important influence on the formation and the scale of debris flows (Shieh et al. 2009). Therefore, the precipitation data before the outbreak of debris flows was collected from local meteorological bureaus, and used as one of the main influence factors to assess the susceptibility of debris flows in this study. In this work, the antecedent precipitation is classified into three categories: inadequacy, medium and adequacy. The classification criteria are listed in Table 1.*" (Page 5, Lines 127-131)

*Table 1 Qualitative grading criteria of antecedent precipitation*

| *Classification* | *Standard of classification* |
| --- | --- |
| *Inadequacy* | *There is no antecedent precipitation or very little antecedent precipitation, which is not enough to make the surface soil moist.* |
| *Medium* | *The antecedent precipitation is intermittent or less, the soil is wet or muddy.* |
| *Adequacy* | *The precipitation lasts for several days, and the soil layer is full of water. Water accumulated in some low-lying areas, and the drainage is not smooth.* |

**4. The results of the field tests mentioned in Section 3.2 should be provided and discussed.**

Answer: Thank you for this suggestion. The results are provided and discussed as follows:

"*Bulk density tests and soil screening tests are carried out in the 70 debris flow deposit areas. Figure 3 shows the results of the bulk density tests. The bulk densities of the soil material in the debris flow deposits are mainly between 1.3 g/cm³-1.8 g/cm³, and the average bulk density is about 1.48 g/cm³. The results of the screening test show that the material composition in the deposit zone is mainly composed of block gravel mixed soil, the content of the block gravel is 30-50%, the content of silt and clay is about 20-40%, and the rest of the deposit material is breccia. The reason for the high content of coarse stone soil is that the collapse phenomenon is quite common due to the active crustal movement in the study area.*" (Pages 6, Lines 133-139)

[Figure]

*Fig.3 Density characteristics of the debris flow deposit in the study area.*

**5. In the Section 5.4, 10 debris flow gullies in the Kaka basin were analyzed to verify the accuracy of the prediction model. Please analyze the reasons why the prediction result of the Linong Gully does not match the actual susceptibility.**

Answer: Thank you for this comment. We add some discussion in the manuscript:

"*Figure 8 shows the catchment of the Linong Gully. The total area of the catchment is about 10.09 km2, and the total amount of loose material is about 4.04×10⁶ m³. The soil material, as shown in Figure 9, is mainly composed of block and crushed stone. Their particle sizes are generally 10-40 cm. In the calculation process, the catchment area is quite large, and then the loose material per catchment area is relatively very small, as shown in Figure 8. Based on the data, the prediction susceptibility of the Linong gully is 2.421, which is very close to the high susceptibility threshold value 2.5. Therefore, although there is a minor deviation, it can still be concluded that the proposed model can perform well to predict the debris flow susceptibility in Southwest China.*" (Pages 10-11, Lines 241-247)

[Figure]

Fig. 8 Distribution of loose deposits of Linong gully

Fig. 9 Soil material in the Linong Gully deposit

**6. The quality of Figure 2 and 3 should be improved. For example, the horizontal axis is not correct.**

Answer: We have revised these figures according to this comment.

[Figure]

Fig.4 Comparison of actual susceptibility and predicted actual susceptibility.

[Figure]

Fig.5 Residual distribution in the regression model of debris flow susceptibility.

---

## Author Comment (AC2) · 11 Mar 2020

**Response to the comments of Anonymous Referee #2 for nhess-2019-349**

**Answers to Technical items for which revision is required** ⋯ '*A multivariate statistical method for susceptibility analysis of the debris flow in Southwest China*'

The authors are grateful for the reviewer's comments and suggestions. The manuscript has been revised and each point of the reviewer's comments has been incorporated and addressed. Your comments have greatly improved the quality of this paper and we hope the revised manuscript will be of suitable standard to be accepted for publication in your journal. The main corrections in the manuscript and the responses to the reviewer's comments are as follows:

**This manuscript presents a multivariate statistical model to predict the susceptibility of the debris flows in Southwest China. According to the topography and geomorphology characteristics in the study area, nine indexes were used to construct a factor index system of the statistical model. Then, 70 typical debris flow gullies in the study area were investigated as statistical samples to generate the model. 10 debris flow gullies on the upstream of the Dadu River were analyzed to verify the reliability of the statistical model. The results showed that the model has a satisfied prediction accuracy. In general, the topic of the manuscript is interesting, the methodology, results and conclusions are presented in a clear way. I recommend publication of the paper after addressing the following comments.**

Answer: Thank you very much for your positive comment on our research. We have studied your comments carefully and made corresponding revisions as required.

**1. Section 2 describes the study areas of this work. Some pictures of the typical debris flow in this area are suggested to be provided.**

Answer: Thank you for your suggestion. Some pictures of the typical debris flows in the study area are provided in Figure 2.

[Figure]

*Fig.2 Typical debris flows in the study area. a) Morphology of the Xianwei Gully along the Yalong River; b) Moraine at the source of the Jiuzhui gully along the Brahmaputra; c) Loose material in the Jiaer gully along the Yalong River; d) Gravelly*

*soil mixed with boulder in Sezu gully along the Dadu River.*

**2. As stated in the Methodology section, the authors carried out a series of bulk density tests, screening tests, drilling and geophysical prospecting. This part should be described in more detail, and the results of these tests should be provided.**

Answer: We agree with this comment. This part is described in detail.

[revised manuscript text omitted]

**3. Line 178-180: how was the criterion determined? why 1.5?**

Answer: The thresholds 1.5 and 2.5 are determined based on the statistical analysis on the debris flows occurred in Southwest China. It is found that the debris flows are very activity when the susceptibility value is larger than 2.5, and the little debris flow occurs when the susceptibility value is lower than 1.5. We explain this point in the manuscript.

"*In Eq. 5, Y is the susceptibility for the debris flow, and the meanings of $x_{11}$, $x_{12}$, $x_{13}$ and other indexes are detailed in Table 4. Based on the statistical analysis on the debris flows occurred in Southwest China, the susceptibility values are classified into three categories in the proposed model:*

$$\begin{cases} Y < 1.5 & \text{Low susceptibility} \\ 1.5 \leq Y < 2.5 & \text{Medium susceptibility} \\ Y \leq 2.5 & \text{High susceptibility} \end{cases} \quad (6)"$$

(Page 9, Lines 202-204)

**4. Line 188-189: How to obtain the "actual values"? Please give more detail.**

Answer: Thank you for this comment. We replace the term "*actual values*" with "*actual susceptibility*" in the manuscript (Line 213). The actual susceptibility of debris flows could be determined based on the "Engineering investigation code for debris flow prevention and control of China" based on the debris flow frequency in the study area.

**5. Section 5 shows the validation of the model. This part should be presented in detail. For example, what does the $R^2$ mean in Table 5? How to calculate the self-test coincidence rate in Table 6? How to define the residual error in Figure 3?**

Answer: Thank you for this comment. We add some explanations in the manuscript.

"*$R^2$ is the fitting degree, which is widely used to evaluate the accuracy of prediction models. As shown in the Table 7, the fitting degree of the proposed model is 71.8%, which shows that this model can precisely predict the susceptibility of debris flows in Southwest China.*" (Page 9, Lines 208-210)

"*In this study, self-test coincidence rate is defined as the ratio of the predicted result to the actual susceptibility. As shown in Fig. 4, the predicted values of debris-flow susceptibility are graded. For the calculated results listed in Table 8, the prediction accuracy for the low susceptibility, medium susceptibility, and high susceptibility debris flows are 78.5%, 92.3%, 82.0%, respectively, which indicates that the proposed model can predict the debris-flow susceptibility well.*" (Page 9, Lines 214-218)

"*Residual error is the difference between a group of values observed and their arithmetical mean. As shown in Figure 5, the residual error of the model mainly fluctuates between ± 0.45, which indicates that the regression line can fit the field value well, and the residual frequency is approximately close to the normal distribution.*" (Page 9, Lines 220-223)

**6. The susceptibilities of 10 typical debris flow gullies on the upstream of the Dadu River are calculated to verify the proposed statistical model. It is suggested to show some pictures about these debris flow sites.**

Answer: Thank you for your suggestion. We add some pictures of the debris flow gullies in the manuscript.

[Figure]

Fig.7 Typical characteristics of Danba section in upper reaches of Dadu River. a) Geomorphology of Bawang Gully; b) Loose deposits in the Mueryue Gully; c) Loose deposits on the trench bed of Shuikazi Gully; d) Abundant source material in the Qiongshan Gully.

**7. The Fig.4 should be more informative, or it is suggested to be combined with Fig.5. Besides, the quality of the figures should be improved, such as Figure 2, Figure 3, and Figure 4.**

Answer: According to this comment, we improve the Figure 2, Figure 3 and combine Figure 4 with Figure 5.

[Figure]

Fig.4 Comparison of actual susceptibility and predicted actual susceptibility.

[Figure]

Fig.5 Residual distribution in the regression model of debris flow susceptibility.

[Figure]

Fig.6 Distribution of debris flow gullies in Dadu river basin.

---

## Author Comment (AC3) · 11 Mar 2020

**Response to the comments of Anonymous Referee #3 for nhess-2019-349**

**Answers to Technical items for which revision is required** ··· '*A multivariate statistical method for susceptibility analysis of the debris flow in Southwest China*'

The authors are grateful for the reviewer's comments and suggestions. The manuscript has been revised and each point of the reviewer's comments has been incorporated and addressed. Your comments have greatly improved the quality of this paper and we hope the revised manuscript will be of suitable standard to be accepted for publication in your journal. The main corrections in the manuscript and the responses to the reviewer's comments are as follows:

**This paper introduces an empirical model for the susceptibility prediction of debris flows in Southwest China. Nine indexes are chosen to construct a factor index system and to evaluate the susceptibility of debris flow. With the model, 70 typical debris flow gullies distributed along the Brahmaputra River, Nujiang River, Yalong River, Dadu River, and Ming River as statistical samples are assessed respectively. 10 debris flow gullies on the upstream of the Dadu River are applied to verify the reliability of the proposed model which suggest a high accuracy of 90% for the statistical model. The paper is general well organized and based on plenty of investigated information. However, there are still many unclear and inexact expressions.**

Answer: Thank you very much for your positive comment on our research. We have studied your comments carefully and made corresponding revisions as required.

**1. In abstract line 14-15, the statement "At present, the susceptibility analysis models of the debris flow in Southwest China is mainly based on qualitative methods. Little quantitative prediction model is found in the literatures." is not true. There have been many research works in the area after "the shock". The author should refer and comment the previous study objectively.**

Answer: Thank you for this comment. We delete this sentence in the abstract and refer the previous study objectively in the Introduction section.

"*In the literature, many models for the debris flow risk prediction in this area have been proposed. For example, Xu et al. (2012) assess the debris flow susceptibility based on information value model and Geographic Information System (GIS) in Sichuan, China. Wang et al. (2016) adopted a self-organizing map method to analyze the susceptibilities of debris flows at the Wudongde Damsite in southwest China. Li et al. (2017) carried out a susceptibility analysis on debris flows also in the Wudongde dam area using the fuzzy C-means algorithm (FCM). Recently, Liu et al. (2018) presented a comprehensive risk assessment model based on semi-quantitative methods to quantify the risk level of each zone in Southwest China. Di et al. (2019) developed a gradient boosting machine (GBM) to predict the susceptibilities of debris flows in Southwest China. Wu et al. (2020) implemented logistical regression models to identify the areas that are susceptible to debris flow formations in Sichuan Province, China. Through the above researches, some promising results have been achieved concerning the susceptibility analysis of the debris flows in Southwest China.*" (Page 3, Lines 68-78)

*Wang, Q., Kong, Y., Zhang, W., Chen, J., Xu, P., Li, H., Xue, y., Yuan, X., Zhan J., Zhu, Y. Regional debris flow susceptibility analysis based on principal component analysis and self-organizing map: a case study in Southwest China. Arabian Journal of Geosciences, 9(18), 718, 2016.*

*Li, Y., Wang, H., Chen, J., Shang, Y.: Debris flow susceptibility assessment in the Wudongde Dam area, China based on rock engineering system and fuzzy C-means algorithm. Water, 9(9), 669, 2017.*

*Liu, G., Dai, E., Xu, X., Wu, W., Xiang, A.: Quantitative assessment of regional debris-flow risk: a case study in Southwest China. Sustainability, 10(7), 2223, 2018.*

*Di, B. F., Zhang, H. Y., Liu, Y. Y., Li, J. R., Chen, N. S., Stamatopoulos, C.A., Luo, Y.Z., Zhan, Y.: Assessing Susceptibility of Debris Flow in Southwest China Using Gradient Boosting Machine. Scientific Reports, 9: 12532, 2019.*

*Wu, S., Chen, J., Xu, C., Zhou, W., Yao, L., Yue, W., Cui, Z.: Susceptibility Assessments and Validations of Debris-Flow Events in Meizoseismal Areas: Case Study in China's Longxi River Watershed. Natural Hazards Review, 21(1), 05019005, 2020.*

**2. In abstract line 21, what is "the quantification theory type I", it never explained in the content.**

Answer: Thank you for the comment. We add some explanations about "the quantification theory type I" in the manuscript:

"*Hayashi's quantification theory is a well-known multivariate statistical method developed by Hayashi (1961). The quantification theory type I applies multiple linear regression methods, which can simultaneously process qualitative and quantitative variables, and evaluate the weight of each variable. Therefore, it is widely used in various fields (Matsumura 2004; Ishihara et al. 2008; Inoue et al. 2009; Shen and Chen, 2018). In this method, the qualitative and quantitative variables could be mutually transformed based on a reasonable principle. Therefore, this method has very good applicability to process the quantitative and qualitative influencing factors of debris flow risk.*" (Pages 6-7, Lines 152-158)

*Matsumura, T.: Analysis of ovipositional environment using Quantification Theory Type I: the case of the butterfly, Luehdorfia puziloi inexpecta (Papilionidae). Journal of Insect Conservation, 8(1), 59–67, 2004.*

*Ishihara, S., Nagamachi, M., Ishihara, K.: Analyzing Kansei and design elements relations with PLS. In 10th QMOD Conference. Quality Management and Organiqatinal Development. Our Dreams of Excellence; 18–20 June; 2007 in Helsingborg; Sweden (No. 026). Linköping University Electronic Press.*

*Inoue H., Tabata H., Tsuji H.: Emotion color combination models using the quantification theory type I and its application to uniform color combination. Transactions of Japan Society of Kansei Engineering, 8(3): 775–781, 2009. (in Japanese)*

*Shen KS, Chen, KH.: Exploring the Critical Appeal of Mobility-Augmented Reality Games. In: International Conference on Kansei Engineering & Emotion Research. Springer, Singapore, 451–459, 2018.*

**3. Line 18-19 and those then after, "longitudinal grade" and "valley slope orientation" are not exact the meaning, maybe "longitudinal gradient" and "valley orientation"**

Answer: We agree with this comment. The term "*longitudinal grade*" is replaced by "*longitudinal gradient*", and the term "*valley slope orientation*" is replaced by "*valley orientation*" in the manuscript.

**4. Line 77, "Study area characteristics of debris flow", not clearly expressed, maybe "Characteristics of the debris flows in the study area"**

Answer: We agree with this comment and replaced "*Study area characteristics of debris flow*" with "*Characteristics of the debris flows in the study area*" in the manuscript. (Page 4, Line 86)

**5. Line 122, in 3.3, it just mentioned "Considerable resources are invested in drilling and geophysical prospecting", but there is no any more information and results provided.**

Answer: According to this comment, the information and results of the drilling and geophysical prospecting are provided in the manuscript:

[revised manuscript text omitted]

**6. Line 178-180 and 190-193, same meaning reappears in very close distance, the sentence is also tedious.**

Answer: According to this comment, we delete Lines 190-193, and modify the sentence in Lines 178-180.

*"Based on the statistical analysis on the debris flows occurred in Southwest China, the susceptibility values are classified into three categories in the proposed model:*

$$\begin{cases} Y < 1.5 & \text{Low susceptibility} \\ 1.5 \leq Y < 2.5 & \text{Medium susceptibility} \\ Y \leq 2.5 & \text{High susceptibility} \end{cases} \qquad (6)"$$

(Page 9, Lines 202-204)

**7. Line 209, "trend" is not a professional expression, should be "strike".**

Answer: We agree with this comment. *"trend"* is replaced by *"strike"* in the manuscript. (Page 10, Line 231)

---

## Author Response (AR1)

**Response to the comments of Reviewers for nhess-2019-349**

Answers to Technical items for which revision is required --- 'A multivariate statistical method for susceptibility analysis of the debris flow in Southwest China'

The authors are grateful for the reviewers' comments and suggestions. The manuscript has been revised and each point of the reviewers' comments has been incorporated and addressed. Your comments have greatly improved the quality of this paper and we hope the revised manuscript will be of suitable standard to be accepted for publication in your journal. The main corrections in the manuscript and the responses to the reviewer's comments are as follows:

**Reviewer #1**

1. Geological drilling was conducted in the active debris flow gullies. The detail information about the drillings conducted in this work should be provided, such as the drilling location, the drilling results.

Answer: Thanks for your suggestion. To find out the material composition and the thickness of the deposit area, the geological drilling was conducted in the active debris flow gullies along the Dadu River, Yalong River, Yaluzangbo River, and Minjiang River. The drilling information and the corresponding soil characteristics are provided in the manuscript:

"The geologic condition in the active debris flow gullies in Southwest China is very complicated. To investigate the material composition and the thickness of the deposit area, the geological drilling was conducted in the active debris flow gullies along the Dadu River, Yalong River, Yaluzangbo River, and Minjiang River. The drilling information, such as the drilling location, drilling depth, and the soil characteristics are provided in Table 2."

Table 2 Information and results of the geological drilling in the study area

| No. | River           | Debirs
flow gully | Geological
coordinates | Drilling
depth (m) | Soil characteristics                                                                                                                                                                            |  |
|-----|-----------------|----------------------|---------------------------|-----------------------|-------------------------------------------------------------------------------------------------------------------------------------------------------------------------------------------------|--|
| 1   | Yalong
River | Reshui
Gully      | 101°16'42"E
28°24'08"N | 15                    | The lithology is mainly metamorphic sandstone and                                                                                                                                               |  |
| 2   | Yalong
River | Reshui
Gully      | 101°16'44"E
28°24'10"N | 22                    | percentage of boulder and gravel is about 40%, which is
slightly angular. Their particle sizes are 40-60cm and 4-9cm,                                                                        |  |
| 3   | Yalong
River | Reshui
Gully      | 101°16'45"E
28°24'12"N | 26                    | respectively. The rest material is silty clay with medium dense.
The cementation state of the soil material in this area is good.                                                            |  |
| 4   | Yalong
River | Shangtian
Gully   | 101°16'26"E
28°24'08"N | 21                    | The lithology is gravel soil with medium dense. The percentage of gravel and coarse sand are 43% and 20%, and the rest of the                                                                   |  |
| 5   | Yalong
River | Shangtian
Gully   | 101°16'29"E
28°24'11"N | 17                    | material is clay. The average thickness of the deposit in this area is about 19.0m.                                                                                                             |  |
| 6   | Dadu
River   | Shuikazi
Gully    | 101°52'07"E
31°03'38"N | 31                    | The thickness of upper layer of the deposit is about 1.5 m, and
the material is weak cemented silty clay with a small amount of
gravel. The thickness of middle layer is about 2.0 m, the |  |

| 7  | Dadu
River              | Shuikazi
Gully  | 101°52'09"E
31°03'39"N  | 36 | material is clay mixed with gravel, containing a small amount
of boulder. The particle size of the gravel, breccia, and boulder
are 2-3 cm, 10 cm, and 40 cm, respectively. The soil content in                                     |
|----|----------------------------|--------------------|----------------------------|----|-------------------------------------------------------------------------------------------------------------------------------------------------------------------------------------------------------------------------------------------|
| 8  | Dadu
River              | Shuikazi
Gully  | 101°52'11"E
31°03'41"N  | 35 | this layer is up to 70%. The lower layer is mainly composed of
gravel and sand, and the particle size is relatively uniform,
generally 5-8 cm. The roundness of the particles is good, and
the content of fine particles is low. |
| 9  | Dadu
River              | Kaka Gully         | 101°52'12"E
31°00'11"N  | 21 | The lithology is mainly mica quartz schist, which is slightly
angular, grayish yellow, dry, and medium dense. The particle                                                                                                             |
| 10 | Dadu
River              | Kaka Gully         | 101°52'14"'E
31°00'15"N | 19 | boulder layer in this gully is mainly filled with silt and a small amount of gravel.                                                                                                                                                      |
| 11 | Yarlung
Zangbo
River | Menda
Gully     | 92°25'12"E
29°15'22"N   | 22 | The deposit in this area is mainly composed of gravelly soil                                                                                                                                                                              |
| 12 | Yarlung
Zangbo
River | Menda
Gully     | 92°25'11"E
29°15'23"N   | 26 | mixed with boulder. The average particle size of the gravels is
15-20 cm, accounting for about 40%. The average particle size
of block stone is about 40-60 cm, accounting for about 10%-                                    |
| 13 | Yarlung
Zangbo
River | Menda
Gully     | 92°25'13"E
29°15'24"N   | 29 | average particle size of 3-4m.                                                                                                                                                                                                            |
| 14 | Yarlung
Zangbo
River | Zhuangnan
Gully | 92°24'23"E
29°15'39"N   | 16 | The material is mainly composed of dense gravelly soil and a                                                                                                                                                                              |
| 15 | Yarlung
Zangbo
River | Zhuangnan
Gully | 92°24'24"E
29°15'41"N   | 11 | of 30-60 cm account for about 30%. The gravels with the
average particle size of 15 cm account for about 10%. The rest                                                                                                                 |
| 16 | Yarlung
Zangbo
River | Zhuangnan
Gully | 92°24'21"E
29°15'42"N   | 17 | obvious miscellaneous accumulation characteristics.                                                                                                                                                                                       |
| 17 | Minjiang
River          | Banzi
Gully     | 103°31'49"E
31°24'25"N  | 18 | The deposit in this area is mainly composed of brown yellow gravel soil, which contains 10% cobble, 45% gravels, and 20%                                                                                                                  |
| 18 | Minjiang
River          | Banzi
Gully     | 103°31'51"E
31°24'27"N  | 24 | coarse sand, and the rest is clay.                                                                                                                                                                                                        |
| 19 | Minjiang
River          | Chutou
Gully    | 103°29'12"'E
31°20'21"N | 14 |                                                                                                                                                                                                                                           |
| 20 | Minjiang
River          | Chutou
Gully    | 103°29'13"E
31°20'22"N  | 17 | The deposit zone in this area is 150 m long and 100 m wide.
The soil material is medium dense, which contains 30%
boulder and 70% gravelly soil.                                                                                    |
| 21 | Minjiang
River          | Chutou
Gully    | 103°29'14"E
31°20'25"N  | 13 |                                                                                                                                                                                                                                           |

2. Table 2 lists nine assessment indexes used in the proposed statistical model. The reason why to select these indexes to evaluate the susceptibility of debris flow gullies should be clarified.

Answer: We agree with this comment. The reason to select these nine indexes has been explained in the manuscript as:

"There are many factors that affect the debris flow formation and development. From the perspective of source material of the debris flows, the main influence factors are catchment area, loose material position and loose material reserves. The antecedent precipitation and H1p rainfall intensity are the main generate conditions of debris flows. Besides, the catchment morphology, longitudinal gradient, average gradient of slope on both sides of the gully, and valley orientation are the main factors to affect the development of debris flows. Therefore, the above nine indexes (listed in Table 3) are selected in this study to assess the susceptibility of debris flows."

**3. In Table 2, the value of antecedent precipitation x83 should be "Fully" rather than "Middle". How to define the antecedent precipitation is "Inadequacy, Middle, or Fully"?**

Answer: Thank you for this comment. The classification standard of the antecedent precipitation is explained in the manuscript as:

"The antecedent precipitation can reduce the soil shear strength, and has an important influence on the formation and the scale of debris flows (Shieh et al. 2009). Therefore, the precipitation data before the outbreak of debris flows was collected from local meteorological bureaus, and used as one of the main influence factors to assess the susceptibility of debris flows in this study. In this work, the antecedent precipitation is classified into three categories: inadequacy, medium and adequacy. The classification criteria are listed in Table 1."

| Classification | Standard of classification                                                                                                                                |  |  |  |  |
|----------------|-----------------------------------------------------------------------------------------------------------------------------------------------------------|--|--|--|--|
| Inadequacy     | There is no antecedent precipitation or very little antecedent precipitation, which is not enough to make the surface soil moist.                         |  |  |  |  |
| Medium         | The antecedent precipitation is intermittent or less, the soil is wet or muddy.                                                                           |  |  |  |  |
| Adequacy       | The precipitation lasts for several days, and the soil layer is full of water. Water accumulated in some low-lying areas, and the drainage is not smooth. |  |  |  |  |

**Table 1 Qualitative grading criteria of antecedent precipitation**

**4. The results of the field tests mentioned in Section 3.2 should be provided and discussed.**

Answer: Thank you for this suggestion. The results are provided and discussed as follows:

"Bulk density tests and soil screening tests are carried out in the 70 debris flow deposit areas. Figure 3 shows the results of the bulk density tests. The bulk densities of the soil material in the debris flow deposits are mainly between 1.3 g/cm3-1.8 g/cm3, and the average bulk density is about 1.48 g/cm3. The results of the screening test show that the material composition in the deposit zone is mainly composed of block gravel mixed soil, the content of the block gravel is 30-50%, the content of silt and clay is about 20-40%, and the rest of the deposit material is breccia. The reason for the high content of coarse stone soil is that the collapse phenomenon is quite common due to the active crustal movement in the study area."

Fig.3 Density characteristics of the debris flow deposit in the study area.

5. In the Section 5.4, 10 debris flow gullies in the Kaka basin were analyzed to verify the accuracy of the prediction model. Please analyze the reasons why the prediction result of the Linong Gully does not match the actual susceptibility.

Answer: Thank you for this comment. We add some discussion in the manuscript:

"Figure 8 shows the catchment of the Linong Gully. The total area of the catchment is about 10.09 km2, and the total amount of loose material is about  $4.04 \times 10^6$  m3. The soil material, as shown in Figure 9, is mainly composed of block and crushed stone. Their particle sizes are generally 10-40 cm. In the calculation process, the catchment area is quite large, and then the loose material per catchment area is relatively very small, as shown in Figure 8. Based on the data, the prediction susceptibility of the Linong gully is 2.421, which is very close to the high susceptibility threshold value 2.5. Therefore, although there is a minor deviation, it can still be concluded that the proposed model can perform well to predict the debris flow susceptibility in Southwest China."

Fig. 8 Distribution of loose deposits of Linong gully